# A Blockchain-Based Spatial Crowdsourcing System for Spatial Information Collection Using a Reward Distribution

**DOI:** 10.3390/s21155146

**Published:** 2021-07-29

**Authors:** Masoud Kamali, Mohammad Reza Malek, Sara Saeedi, Steve Liang

**Affiliations:** 1Department of GIS, Faculty of Geodesy and Geomatics Engineering, K. N. Toosi University of Technology, Tehran 19967-15433, Iran; mrmalek@kntu.ac.ir; 2Department of Geomatics Engineering, University of Calgary, Calgary, AB T2N 1N4, Canada; steve.liang@ucalgary.ca

**Keywords:** blockchain, location-based services, crowdsourcing, reward algorithm

## Abstract

Due to the increasing relevance of spatial information in different aspects of location-based services, various methods are used to collect this information. The use of crowdsourcing due to plurality and distribution is a remarkable strategy for collecting information, especially spatial information. Crowdsourcing can have a substantial effect on increasing the accuracy of data. However, many centralized crowdsourcing systems lack security and transparency due to a trusted party’s existence. With the emergence of blockchain technology, there has been an increase in security, transparency, and traceability in spatial crowdsourcing systems. In this paper, we propose a blockchain-based spatial crowdsourcing system in which workers confirm or reject the accuracy of tasks. Tasks are reports submitted by requesters to the system; a report comprises type and location. To our best knowledge, the proposed system is the first system that all participants receive rewards. This system considers spatial and non-spatial reward factors to encourage users’ participation in collecting accurate spatial information. Privacy preservation and security of spatial information are considered in the system. We also evaluated the system efficiency. According to the experiment results, using the proposed system, information accuracy increased by 40%, and the minimum time for reviewing reports by facilities reduced by 30%. Moreover, we compared the proposed system with the current centralized and distributed crowdsourcing systems. This comparison shows that, although our proposed system omits the user’s history to preserve privacy, it considers a consensus-based approach to guarantee submitted reports’ accuracy. The proposed system also has a reward mechanism to encourage more participation.

## 1. Introduction

The term crowdsourcing was coined in 2006, with the concept of using people to perform a task in the form of an open call [1]. Applying this concept and using the hired population’s location to perform specific tasks creates a concept called spatial crowdsourcing [2]. The relevance of sensitive information in spatial crowdsourcing systems, especially spatial information, has led to the use of various encryption methods and technologies to store this information. With the advent of blockchain technology by Satoshi Nakamoto in 2008 [3], there has been widespread use of this technology in different fields. The characteristics of blockchain, such as distribution, transparency, security, and immutability, increased its popularity in various applications, such as finance [4,5], integrity verification [6,7,8], governance [9,10], internet of things [11,12,13], healthcare management [14,15,16], privacy and security [17,18], and education [19,20,21]. Besides, different platforms are designed to implement blockchain technology in various aspects [22,23,24]. In addition, the use of blockchain in crowdsourcing preserves the users’ privacy [25,26], secures crowdsourcing systems, and makes transactions between users transparent. Generally, the central authority in the crowdsourcing system endangers the users’ privacy and reduces security and transparency [27,28]. Additionally, to the best of our knowledge, there is no rewarding mechanism for requesters in traditional crowdsourcing systems [29]. Moreover, collecting accurate spatial information can be effective in better allocating resources. In this paper, based on the use of blockchain technology in spatial crowdsourcing, we proposed a system to increase the quality of the information received from public reports in terms of transparency, security, and accuracy. Due to the lack of proper validation methods in crowdsourcing systems and the relevance of this issue in better information management and resource allocation, this article aims to establish a secure system to prevent entering incorrect information deliberately or inadvertently by users of the system. Consequently, the spatial information sent by each user must be verified by other users at that location to record the report in the blockchain-based spatial crowdsourcing system. In other words, the received report is initially provided as a task to other users in the target location. In the proposed system, a user logs into the crowdsourcing application and submits a location-based report. Afterward, other users are selected based on the location of the report as workers to confirm or reject the report as a task. Once confirmed, the system records the information as a transaction in the pending list of the designed private blockchain. In summary, the main contributions of this study are as follows:We proposed a blockchain-based system that increases transparency and security in the system compared to centralized systems.The proposed system uses crowdsourcing to collect spatial information in the form of reports, which reduces the entry of incorrect information into the system. The impact of this reduction on better resource allocation is also examined.To our best knowledge, this is the first system to reward all users based on spatial and non-spatial factors. We also provided formulas for calculating this reward.Moreover, due to the relevance of spatial information in the system, we performed user privacy and security analysis.

The rest of this article is as follows. Section 2 presents a summary of related work in the field of crowdsourcing and the use of the blockchain technology in this area. Section 3 explains the problem and introduces the architecture of the proposed system; it also states the threat model and goal of the proposed system. In Section 4, we implement the proposed system. Section 5 presents security and privacy analysis. In Section 6, we evaluate the system performance. Finally, Section 7 concludes this work and outlines future works.

## 2. Related Works

Recently, different words, which have the same meaning, have been used to introduce the concept of spatial crowdsourcing [30]. Terms such as place-centric crowdsourcing [31,32], mobile crowdsourcing [33,34,35], and location-based crowdsourcing [36,37,38,39] are all used to perform a task depending on the user’s location. Different categories are provided for spatial crowdsourcing applications [30]. Applying spatial crowdsourcing to gathering information in a specific location [40,41,42,43,44,45,46], answering spatial queries, and using local people’s knowledge to solve problems within a particular area are applications of this concept [47,48,49,50]. In addition, spatial crowdsourcing is used in location-based services to detect traffic incident [51].

The quality of data collected by individuals should be evaluated from various aspects [52,53,54]. The accuracy, precision, and security of the information collected in spatial crowdsourcing are among the qualitative elements. In addition, some articles have highlighted the importance of privacy-preservation in spatial crowdsourcing systems [55,56].

According to the different architectures used in the design of crowdsourcing systems, we have classified them into three. The first category is centralized crowdsourcing systems [57,58,59,60,61]. The assumption of the centralized systems is to have a third trust party for storing information and performing other crowdsourcing processes. A centralized crowdsourcing system often suffers from a lack of security and privacy, and since there is sensitive information in spatial crowdsourcing systems, they are not good choices. The second category is decentralized crowdsourcing systems. These systems are designed to solve problems of centralized systems. Zhang et al. have developed a distributed dissemination protocol and minimized the makespan by designing a delay tolerant network [62]. There has been a study on using social relationship factors for task assignment in the crowdsourcing system [63]. Cheung et al. in [64] provided an algorithm for distributed and anonymous task solutions that improve computing speed in crowdsourcing systems.

Although the creation of these systems could solve many security and privacy issues, the lack of traceability and transparency in how users perform their tasks led to another category of architecture called distributed crowdsourcing systems. Recently, various crowdsourcing systems used blockchain platforms in their design [65]. By integrating the concept of crowdsourcing and blockchain, a framework has been developed to improve the judiciary process by using collective intelligence, which has led to increased transparency in judgments. A crowdfunding system, called Betfunding, is based on a blockchain that enhances the security and transparency of users’ financial transactions. This system classifies users into three groups: (1) funders who are interested in project development but lack sufficient knowledge or resources; (2) developers who possess enough knowledge of project development; and (3) judges who play the role of a trusted third party in the system. The Solidity programming language has been used to implement smart contracts on Ethereum networks [66]. In addition, the use of blockchain in crowdfunding systems helps regulators manage investor money and solve financial security problems. The use of blockchain in social networks has also been of interest. Given the relevance of using a trusted structure to assign tasks to workers in crowdsourcing, blockchain-based crowdsourcing systems have been designed and implemented. Some systems have used cryptocurrencies to incentivize users. crowdBC is a blockchain-based framework for crowdsourcing that is implemented on the official Ethereum test network [67]. There are four leading roles in this framework—requester, worker, CrowdBC client, and miner. First, a requester sends task description and reward amount to the system. Then, the system selects workers based on their skills and best bid, which is done locally in the user’s system by the crowdBC client. Finally, a miner adds previous transactions into the blocks and provides security for the system [28]. A private and anonymous decentralized crowdsourcing system called ZebraLancer focuses on two critical issues—data leakage and identity breach [68]. In this system, task results are encrypted using the requester’s public key and are placed in a smart contract. The results are then evaluated according to the instructions provided by the requester on the blockchain. If a worker submits a task twice, no one can link to it. This system has been implemented in the Ethereum smart contract. The combination of crowdsourcing and blockchain is used in various areas. Improving the security of transportation systems has led to the development of a blockchain-based intelligent transportation system framework [69]. In addition, to encourage citizens to report problems in smart cities, a secure distributed crowdsourcing network called Vizsafe has been implemented; it uses Ethereum to incentivize people to submit reports [70]. Public participation GIS (PPGIS) is one of the other aspects where blockchain can increase citizens’ participation in decision-making [71]. In this system, smart contracts have been implemented in the Ethereum network, and the web3.js [72] library is used to interact with smart contracts. Blockchain technology is useful in solving real-estate problems. All contracts between users are stored in the blockchain and cannot be changed. Besides, users can apply for investment, mortgage, loans, and other financial assistance. Investors and financial institutions can use the system to search for different offers on financial issues [73]. Designing a crowdsourced energy system (CES) for small-scale collaborative power supply with the help of cryptocurrencies and blockchain is another application of this technology in crowdsourcing systems. Implementing CES on a large scale requires a secure framework, and blockchain and smart contracts are good choices in this area [74]. With the advent of the internet of things, the use of massive sensors to collect information and establish secure communication between sensors has employed the blockchain in this area [75]. Crowdsourcing is an appropriate solution for sharing knowledge that can solve problems faster and cheaper [76]. CPchain is a copyright-preserving crowdsourcing data trading framework based on blockchain. Consumer, seller, IPFS, smart contract, and digital fingerprint-embedding algorithm are the system’s five entities. In this system, the smart contract plays the role of a broker [77]. A mechanism for fine-grained authorization in data crowdsourcing has also been proposed. Therefore, the third-party entities evaluate the data. Cloud storage is used to store encrypted data [78]. TSWCrowd is a decentralized task-select-worker framework that sorts tasks to ensure reliability and prevent completing tasks without payment. It uses smart contracts to publish tasks and select workers for the tasks [79].

Although using blockchain in crowdsourcing systems solves many problems related to security, transparency, traceability, and privacy [27,80,81], the current systems have many issues.

Based on the current literature review, there are several gaps in spatial crowdsourcing systems:To the best of our knowledge, the rewarding mechanism for requesters is overlooked in crowdsourcing systems [29]. Furthermore, the lack of spatial factors to calculate the rewards of various users who participate in spatial crowdsourcing systems poses problems.Using users’ history in crowdsourcing systems reveals the users’ sensitive information and endangers their privacy [80].The lack of secure and transparent distributed spatial crowdsourcing systems for collecting accurate information is a significant issue and less research has been carried out in this area [81].

The proposed system uses a rewarding mechanism for all different roles in the spatial crowdsourcing system to collect accurate spatial information.

## 3. Problem Statement

In this section, we first introduce the spatial crowdsourcing system architecture and then explain the threat model and the assumptions. Finally, we present the notations and goals of this research.

### 3.1. The Proposed System Architecture

In the system architecture, users are classified into three. However, initially, they have no particular distinction and may play different roles at any given time. The three classes are requesters, workers, and miners. In the designed system, requesters submit reports, and workers are assigned only because of their location and without any intervention from other users. Miners are responsible for mining blocks in the blockchain network. Figure 1 shows the system architecture of this blockchain-based spatial crowdsourcing system. In this system, the requesters determine the location and type of these reports. The type of report can be any of urban issue, including inundation and fire. Users do not have a particular role in this system until a report is submitted.

After registering the report, the workers are selected according to their location at that moment. The role of workers is to confirm or reject reports recorded in specific locations. If more than half of the workers confirm the type of report in the desired location, this report will be confirmed and sent to the blockchain. A number is considered for the minimum number of workers confirming the report. All users in this system can be miners, and after confirming the report by the workers, they can mine the blocks in the blockchain. If workers do not confirm the report, it will be removed. The requester is unable to identify the workers, their location, and votes. There are no restrictions on sending reports to the system, and a user can play three different roles in a short time. The requester cannot simultaneously play the role of a worker for the report they submit. Although the workers are selected based on the report’s location, the requester does not need to be in the location where the report is about. In addition, the location of miners is irrelevant in this system.

### 3.2. Notations

Abbreviations section shows all the notations used in this article, some of which are described below for better understanding.

Each *N_i_* can play three main roles in the blockchain-based spatial crowdsourcing system. *W_i_*, *M_i_*, and *R_i_* are the roles that each user *i* can play in the system. *R_i_* submits *RE_i_* to the system. The system specifies *U_w_* for validation of *RE_i_*. *U_w_* votes to confirm or reject *TRE_i_* that is located at *LRE_i_*.*ID_w_* and *ID_r_* are the identifiers of each worker and requester. *CoRE_i_* = (*ID_r_*, *ID_re_*, *TRE_i_*, *T_re_*, and *LRE_i_*) and *LRE_i_* = (*Lat_i,re_*, *Lon_i,re_*), which is compared with each *LW_i_* = (*Lat_i,w_*, *Lon_i,w_*).*T_i_* is placed in *Pen_t_*. We have many *B_i_* in the designed blockchain, and each *B_i_* has numerous *T_i_*. *T_i_* has unique *ID_t_*. *CoRE_i_* is the content of each report, and *CoT_k_* is the content of each transaction in the blockchain-based spatial crowdsourcing system.*PEY_r_* is the reward of requesters when their reports are confirmed and *PEY_m_* is the miners’ reward after mining. *PEY_w_* is the reward of the workers in this system.

### 3.3. Research Assumptions and Threat Model

In this section, we first present the research assumptions. In addition, we examine the effects of malicious users on the system. In the threat model, we analyze the system’s role against the destructive effects of newbies.

#### 3.3.1. Research Assumptions

Nodes cannot record transactions directly in the blockchain and do not have access to unconfirmed reports in the system.Submitting reports by the requesters and assigning tasks to the workers is carried out through a secured communication channel.Reward payments are transferred securely.

#### 3.3.2. Threat Model

Given that the ability and activity of each role are defined in the proposed system, the thread model has been inspired by the Persona Non Grata method to focus more on the potential threats of users and their motivations in different roles [82]. Due to the various roles of users in the proposed system, we have divided the potential threats into five main categories: malicious requesters, malicious workers, malicious miners, newbie requesters, and newbie workers. These categories are described in the following paragraphs.

The first category is malicious requesters. In such crowdsourcing systems, malicious requesters may submit a report several times to receive more rewards (in the designed system, all participants receive a reward). Malicious requesters try to manipulate the information in the confirmed reports before sending them to the blockchain. The requesters may also want to make changes to the registered reports, submit them in their names, manipulate the correct information, and record incorrect information in the system. Requesters may want to receive more rewards by changing the system’s incentive policies. They may submit false reports and ask malicious workers to confirm them to get more rewards, which will lead to an unfair distribution of tasks among the workers. Some malicious requesters may want to submit their reports to the blockchain without confirmation.

The second category is malicious workers. To get more rewards, malicious workers in this system may want to confirm a report several times, or some workers may ally with each other to confirm incorrect reports. Malicious workers may want to confirm a report without being in the location. Some of them try to obtain sensitive information from requesters or other workers to abuse them. They attempt to make changes to incentive policies to get more rewards for each report. Sometimes they try to get a reward without performing their tasks. Workers also reject a valid report so that they can submit it as a requester. Workers try to confirm or reject a report based on the votes of other workers.

The third category is malicious miners. There are malicious miners in a system that tries to create fork chains to get more rewards.

Newbie requesters are the forth category. Some newbie requesters are not intentionally malicious users and do not benefit from their incorrect behavior in the system. Due to insufficient knowledge, they cause harm to themselves and the workers in the system. This behavior can be the wrong choice of report type, which is selected due to a misunderstanding of what happened. In addition, they may submit a previously registered report, which will reduce the system’s performance.

The last category is newbie workers. Some newbie workers are not malicious, and their behavior is not due to abuse of the system and other users. However, due to insufficient knowledge, they may harm requesters or other workers in the system. This behavior can be incorrect confirmation or rejection of reports.

In some centralized and distributed crowdsourcing systems, there is insufficient knowledge about the task created by the requesters and the evaluation of the task performed by workers in the system. They are only responsible for preventing fraud in payments and contracts between the requesters and workers, which is performed by a centralized server or smart contracts in blockchain-based systems. New users may cause problems in these systems.

### 3.4. Design Goals

This blockchain-based spatial crowdsourcing system aims to improve some vulnerabilities in both traditional and current crowdsourcing systems. With the advent of blockchain technology, some problems in centralized systems have been solved. These problems include the existence of a third trusted party to perform all activities related to assigning tasks among workers, rewarding, and confirming workers’ tasks. Using blockchain in crowdsourcing can solve these problems. Despite solving these problems in distributed systems, the lack of requester’s knowledge in creating provisions of the contract and, on the other hand, the lack of awareness of the workers about the intended tasks causes issues in such systems. These problems may be due to users’ different perceptions of a concept. Besides, the lack of monitoring of contracts between requesters and workers causes damage to various nodes in the network.

Moreover, improving the quality of spatial information from the perspective of accuracy can be the solution to some problems in decision-making and resource allocation. The proposed system aims to solve the above-mentioned issues, which are summarized below.

Traditional systems perform all the processes of the spatial crowdsourcing system by a centralized authority, which causes risks such as single-point failure. In these systems, the selection of workers and the confirmation or rejection of tasks submitted by the requester are made by the central authority. In addition, in the case of data leakage, sensitive user information, including spatial information, task solutions, and user rewards, is abused. Besides, these systems use the central authority to determine the distribution of rewards and the confirmation of workers’ tasks that blockchain solves these problems.The existence of transparent participatory systems for the public and the ability to trace all system activities to improve performance is one of the blockchain-based crowdsourcing system purposes.Current data collection systems are inaccurate in data collection, which is more common in location-based systems. Improving the quality of these systems, in terms of accuracy, is another goal of this blockchain-based spatial crowdsourcing system, which is carried out with the confirmation or rejection of reports by workers.Despite the benefits of current distributed systems, they have drawbacks. Insufficient requester knowledge in creating provisions of contract can affect the worker’s confirmation or rejection of a task. In these systems, workers can only receive their rewards if they meet the requesters’ requirements. Besides, there is no evaluation for requesters, which results in an unfair evaluation of workers by requesters. In the proposed system, this problem is solved using the votes of the selected workers for each report.Better management and allocation of resources in different regions due to reduced incorrect information is another goal of the proposed system.

## 4. The Proposed Blockchain-Based Spatial Crowdsourcing System

### 4.1. Overview

By establishing this system, we have provided a secure and transparent environment for gathering spatial information. The system is built to collect spatial information based on the use of crowdsourcing. It uses incentive policies to encourage users to participate in the system. Generally, people do not participate in such systems due to the lack of motivation and privacy. Facilities are responsible for reviewing confirmed reports in the system and resolving related issues. The use of blockchain enables users to evaluate the performance of related facilities. Facilities are also required to provide accurate information to the public. In this system, no role is initially assigned to the users, and they register in the system in the same way. No personal information is required for registration. *R_i_* does not need to be in the location listed in the report and can submit the report from another location. After submitting *RE_i_*, the location of the workers in the system is determined. *U_w_* close to the location of the report are selected. The number of *U_w_* selected for each report must be greater than 2, although this number is optional. If more than half of the specified workers confirm the report, it will be permanently recorded in the blockchain. *RE_i_* is stored as *T_i_* in *Pen_t_*. All users have access to the blockchain and can view the information. In this system, all users are miners. After registering the transactions, *B_i_* is mined, and *T_i_* is placed in *B_i_*. *ID_r_* and *ID_w_* confirmed *RE_i_*, *LRE_i_*, *TRE_i_*, *T_re_* and are the items that are stored in the *T_i_*. *LR_i_* is not recorded in *T_i_*. To motivate users to participate in this system, all users can receive rewards according to the factors considered. Consensus algorithms are used to maintain blockchain integrity. Two main functions are used to design the system’s processes. The first function is task management, which includes registration, task submission, task assignment, and task confirmation. The second function is blockchain management, which determines the structure of blocks and the incentive mechanism. After performing the above steps, the reports recorded in the blockchain are accurate and precise. In addition, they are traceable to the public.

### 4.2. Proposed System Implementation

#### 4.2.1. Task Management

This function includes user registration, task submission, worker selection, and report confirmation. After completing these steps, the reports will be recorded as a transaction in the blockchain.

##### Registration

The only way to enter the system and access the blockchain is to register through the system interface. There is no need for sensitive user information to register in the system. When a user registers in the system, it will use the consensus algorithm to access all the information in the blockchain, including the pending list and the mined blocks.

##### Task Submission

After registration, the user can select *LRE_i_* and *TRE_i_* that are predefined in the system. The user does not need to be in *LRE_i_*, and the *LR_i_* will not be recorded in the system. *CoRE_i_* includes *TRE_i_* and *LRE_i_* that are in the form of *Lat_i,re_* and *Lon_i,re_*, *T_r_*, and *ID_re_*. *R_i_* determines *LRE_i_* on the map so that the coordinates are more accurately recorded.

##### Task Assignment

After submitting the report, the location of other users will be taken. Afterward, the distance between users and the report is obtained. Workers are selected according to the radius considered in the system up to the location of each report. There is no information about workers except for their location. The requesters and other workers also have no information about each other before and after the selection. In addition, to assign tasks, users’ history is not considered for selection. The combination of workers is stored for each report because workers who are close to each other are not always in the same group. Moreover, users in the same area may confirm or reject reports for higher profit, which reduces the system performance. To solve this problem, the combination of workers will change randomly depending on the number of people in the report’s location. With this condition, it is not possible to assign the same group of workers for several reports, and this prevents fraud in assigning tasks. The task assignment step is presented in Algorithm 1. Different algorithms are used to measure distance. In this system, the distance between the reports and workers is calculated on the sphere. The reason for this is the difference between the distances on the sphere and plate. We used the Haversine formula to calculate the distance [83]. This formula uses the location in latitude and longitude coordinates.
(1)θ=d/r
(2)$$\mathrm{hav}\left(\theta \right)\;=\mathrm{hav}\left(\varphi_{2}-\varphi_{1}\right)\;+\cos\left(\varphi_{1}\right)\cos\left(\varphi_{2}\right)\mathrm{hav}\left(\lambda_{2}-\lambda_{1}\right)$$
(3)$$\mathrm{hav}\left(\theta \right)\;=\sin^{2}\left(\theta /2\right)\;=\;\left(1-\cos\left(\theta \right)\right)/2\;d=r\;{\mathrm{hav}}^{-1}\left(h\right)\;=2r\sin^{-1}\left(\sqrt{h}\right)\;=2r\sin^{-1}(\sqrt{\sin^{2}\left(\varphi_{2}-\varphi_{1}/2\right)\;+\cos(\varphi_{1})\cos\left(\varphi_{2}\right)\sin^{2}\left(\lambda_{2}-\lambda_{1}/2\right)})$$
**Algorithm 1.** Task Assignment Algorithm.**Input** Report; Users**Output** Workers1: Get the report’s location2: Calculate the region of the report3: Get users’ location4: Calculate the distance between users and the report5: Get the combination of workers in each report6: **if** UserDistance < ReportRegionDistance **then**7:  WorkersInRegion = Users;8:  **if** WorkersInRegion **in** WrokerCombinationInEachReport **then**9:      **return** False;10:  **else**
11:      Workers = WorkersInRegion;12:  **end if**13:  **return** Workers14: **end if**


##### Task Confirmation

Once workers are determined, they are asked to confirm or reject the report. The location and type of each report are sent to workers. Given the spatial radius, they have time to observe the location of the report and then vote. If more than half of the workers confirm a report, the report is permanently recorded in the blockchain. The task confirmation step is presented in Algorithm 2. Otherwise, the report will be removed after a while, and no information will be available on the blockchain network. When users vote for a report, they do not have information about other workers’ votes in the system before confirmation. Afterward, the ID of the users who have confirmed the report becomes available in the blockchain. In addition, the workers’ votes have the same rank. Consequently, it is only the vote of the group that determines the confirmation or rejection of a report.
**Algorithm 2.** Task Confirmation Algorithm.**Input** Workers; Report**Output** Confirmed report1: Request for confirmation to workers2: Confirm or reject the report3: **if** confirm = True **then**4:  **return** report confirmation;5: **end if**6: **if** NumberOfConfirmation > NumberOfWorkers/2 **then**7:  send report to blockchain;8:  **return** confirmed report;9: **end if**

#### 4.2.2. Blockchain Structure

After the task management is completed, the report is sent to the blockchain in the form of a transaction. A new block is mined, and the transaction is placed into the block. Then, the participants receive their rewards based on specified factors.

##### Transaction Structure

Each transaction includes the transaction ID, the type and location of the report, and the time of its submission (Figure 2). This structure helps to improve the transparency and traceability of the system, and all users can check their performance in the system.

##### Incentive Mechanism

The system’s incentive policy is for all users. After the block is generated, all users who participated get their rewards. A combination of spatial and non-spatial factors is used to determine the value of the user reward in the proposed system. The reward value is different for each user. Spatial factors for determining the amount of the reward include the spatial diversity of the confirmed reports by each requester and the spatial diversity of the reports confirmed by the worker. Non-spatial factors considered include the number of workers that confirmed the report and the coefficient for adjusting the reward amounts, which are different for each role. The reward for miners in this system is the same. In (4), γ is the adjustment coefficient of the reward, which is between 0 and 1, and it is higher for the requesters than workers.


(4)$$PEY_{r}=\gamma \times N_{re}\times \sum_{i,j=0}^{n}d_{ij}$$



(5)$$PEY_{w}=\gamma \times 1/N_{w}\times N_{c}\times \sum_{i,j=0}^{m}d_{ij}$$


$N_{re}$ is the number of approved reports for each requester and is the sum of the distances between this report and the previous reports of the requester. The higher the value, the greater the spatial diversity of the requester’s reports. In (5), *N_c_* is the sum of the reports confirmed by a worker, and $\sum_{i,j=0}^{m}d_{ij}$ is the sum of the intervals between this report and the previous reports confirmed by the worker. This value indicates the spatial diversity of the reports that the worker has confirmed and is recorded in the blockchain. *N_w_* is the number of workers who have confirmed the report. The lower the number, the higher the reward for workers. All factors are calculated from the blockchain.

## 5. Security and Privacy Analysis

In this section, we first discuss the security of the system against some attacks and then explain the users’ privacy in the system.

### 5.1. Security Analysis

Security analysis in this article includes Byzantine fault tolerance, reputation fault tolerance, transaction malleability, time jacking, race attack, distributed denial of service (DDoS) attack, decentralization, and no single-point failure. We also look at how to deal with these attacks on the system.

#### 5.1.1. Byzantine Fault Tolerance

In distributed systems, it is essential to broadcast the correct values between the different nodes of the system. Byzantine faults have negative effects in the crowdsourcing systems [27]. Broadcasting the incorrect information in the system causes incorrect reports in terms of location and type. Furthermore, changing the workers of the confirmed reports in the system causes the same workers to be selected for reports in the same locations, revealing the exact location of the workers. The use of POW consensus algorithm in the blockchain of the spatial crowdsourcing system maintains the integrity of the information in the system, which makes it resistant to Byzantine faults.

#### 5.1.2. Reputation-Based Attacks

The different privileges that users have in systems cause various attacks to increase or decrease the reputation of users. This has some effects on how to calculate rewards and task assignments. In this system, users do not have any privileges, and assigning tasks is based on the location of workers. All miners have equal rights to mine blocks. Requesters have no priority over others when submitting a report. The privileges considered for calculating rewards are spatial and non-spatial factors computed after workers’ selection.

#### 5.1.3. Transaction Malleability

Transactions may change after they are created and before added to the blockchain. These changes are usually applied to transaction IDs. In this system, the immutable elements of each transaction are used, including the location of a report, worker’s ID, and requester’s ID. User rewards are calculated on the basis of these elements, and changing the transaction ID does not disrupt the system.

#### 5.1.4. Time Jacking

In this attack, the timestamp of the target node is manipulated. In this case, the node is isolated and fraudulent transactions are made and sent to the target node. In this system, the node information is integrated with all other nodes. Valid blocks are always placed in all nodes.

#### 5.1.5. Race Attack

In this attack, the adversary makes two transactions—one legitimate and one fraudulent. It sends the legitimate transaction to the mining pool and the fraudulent one to the target node so that it can obtain services from the node without providing a valid transaction. In our system, no user is allowed to add two reports with the same location and type in a certain period. In addition, the calculation of the reward is based on a valid chain. Consequently, the existence of these fraudulent transactions in the system will not affect and will be eliminated.

#### 5.1.6. DDOS Attack

In this attack, the adversary sends a large number of requests to cause problems for different parts of the system. In our system, each request must be passed through the workers’ filter before being registered in the blockchain. If workers confirm the report, it will be registered as a transaction. Consequently, no matter how many requests there are, they are not directly added to the blockchain to disrupt the system. It is only possible that the number of transactions will increase, which due to the time lag in the reports’ confirmation by workers, will not be registered into the system all at once.

#### 5.1.7. Decentralization

There is a possibility that there are powerful miners in blockchain-based systems. These miners try to control the blockchain with more computing power. To solve this problem, the number of blocks that each miner can mine at a given time is constant. If it exceeds this value, the block in the system is considered invalid.

#### 5.1.8. No Single-Point of Failure

Due to the intrinsic characteristics of the blockchain and the role of all users as miners, this problem is solved.

### 5.2. Privacy Analysis

Privacy preservation is an essential factor in spatial crowdsourcing systems. It can increase users’ participation.

#### Pseudonymity

In our system, the real identity of the users is not used for authentication. There is also no location information for requesters in the system. As mentioned, requesters do not need to be in the location of the report, and the report’s location is recorded only in the system. After the report is recorded in the blockchain, there is no information about the location of the workers. Besides, workers may be in any location after the transaction is registered, which preserves the privacy of workers. In addition, owing to the use of cryptocurrencies to pay bonuses, there is no need for users’ banking information, which prevents revealing their real identities.

## 6. Performance Evaluation

This section first explains the system settings in which we implemented the user interface and blockchain network. Then, to evaluate the performance of the proposed system, we consider a scenario in which we examine the number of assigned reports to facilities and the time interval to review them at specified distance intervals. Finally, we provide a comparison between the proposed system and other existing crowdsourcing systems. The five factors considered for this comparison are privacy, worker credential, worker experience, requester reputation, and requester reward.

### 6.1. System Setting

We implemented the system user interface using the JavaScript programming language and React Native Framework on Android and iOS platforms. In addition, we used Node.js Express Framework to implement the blockchain network. We used the sha256 algorithm for hashing blocks in the blockchain.

### 6.2. Resource Allocation

By using the proposed system, the collection of accurate reports is increased which results in better resource allocation and faster review time. To examine the proposed system, we considered a scenario to evaluate the effect of removing incorrect reports on the review time of submitted reports. In this scenario, we assumed that the submitted reports can only be checked by field observations. We first placed 100 points to represent facilities randomly in Charlotte city, North Carolina. The users submitted their reports using the system user interface. We considered 1000 points as reports received for those facilities, which had been randomly selected. Afterward, the workers were selected based on their closeness to the reports’ location.

To calculate the minimum time interval between facilities and reports, we measured the distance between each facility and reports in the area. Then, according to the number of facilities and reports, we allocated several reports to each facility. By calculating the time interval between each facility and the related reports, as well as the distance of reports with each other, we obtained the minimum time interval between each facility and related reports. The number of reports assigned to each facility depends on the location of the reports and facilities. We obtained these distances from the street network in the area. Confirmed reports should be reviewed at the reports’ location. Consequently, the Euclidean distance is not suitable for calculating the time interval and the distance on the street network must be calculated. Figure 3 shows how these facilities are allocated to reports.

To evaluate the impact of the proposed system, we assumed that each worker would confirm or reject the reports, and we considered a number of these reports randomly invalid in terms of location and accuracy of the information. Then, we calculated the minimum distance in different steps. These changes were carried out in four steps. At each step, we considered 10% of the 1000 received reports incorrect. Figure 4 shows that using the proposed system prevents the registration of incorrect information to the system in each step, which reduces the number of reports assigned to facilities in each distance interval. The distance interval refers to the distance between assigned reports and facilities. This reduction will lead to better resource allocation and speed up the review time of accurate reports. Figure 5 also shows that rejecting incorrect reports reduces the time interval to review facilities’ accurate reports in specific distance intervals between facilities and assigned reports.

Table 1 shows the minimum time interval for facilities to review total reports in each step. According to the obtained values, if 40% of the reports are incorrect and rejected by workers, 505.15 min will be reduced from the optimal time interval. This reduction will lead to a faster review of critical reports in the region and the allocation of more resources to future reports in the system.

### 6.3. Comparison

To date, various systems have been created for crowdsourcing. These systems are compared based on different characteristics in Table 2. In this comparison, the characteristics of privacy, worker credential, worker experience, requester reputation, and requester reward are considered.

The first criterion is privacy. Due to the sensitive information in these systems, it is crucial to preserve users’ privacy. Some crowdsourcing systems do not use people’s real identities to participate in the system, which helps with privacy preservation. The most crucial factor in the popularity of a crowdsourcing system is how it increases the quality and quantity of user participation. Privacy preservation increases user participation in spatial crowdsourcing systems. The second criterion in this comparison is worker credential. In some existing systems, worker credential is used to allocate tasks. Supposing that users’ credibility increases with their participation in the system, some users will confirm all reports submitted to them, which will enter the incorrect information into the system. In addition, not using user history leads to preserving the users’ privacy and greater user participation in the crowdsourcing system. The third criterion in this comparison is the experience of the workers. In some systems, workers’ experience in assigning tasks is taken into account. Workers’ experiences are an essential factor in reputation-based attacks. The fourth criterion in this comparison is the reputation of the requesters. This allows requesters to register fake tasks in the system to increase their reputation.

Our proposed system aims at increasing the accuracy of spatial information by a census-based approach. According to the jury theory, a group of fewer competent individuals is more likely to make the correct decision than a more competent individual [87]. For this reason, the proposed system uses the vote of a group of workers instead of user history to confirm or reject reports. In the proposed system, users’ history does not affect the importance of workers’ vote so each worker’s vote is not preferable to the other workers. For this reason, it is not necessary to use the worker credentials, requester reputation, and worker experience.

On the other hand, using user history increases the possibility of various attacks in the system. This history can be the user skill. Forging these skills can cause problems in the system. The unfair distribution of workers is one of these problems. In some systems, requesters select workers for tasks. They also provide filters to eliminate workers and give them rates after they have performed their task. All of this is carried out according to the requesters’ knowledge, and their opinion influences the choice or rating.

The last criterion in this comparison is requester reward. Having a rewarding mechanism for users is one of the crucial factors in increasing the quality of user participation in spatial crowdsourcing systems. Malicious requesters use various tricks to make more benefits and reduce workers’ rewards. In most crowdsourcing systems, requesters create a task, and the workers do it and receive their reward according to the quality of the task performed. These systems do not offer rewards to requesters. In the proposed system, the use of spatial and non-spatial factors to calculate the rewards of requesters encourages users to participate in the system and collect accurate spatial information. Users in this system can also have various roles in different reports, which allows each user to be rewarded in the role of requester, worker, and miner. Motivating requesters to create tasks can improve the overall performance of the system and attract more workers to the system. Besides, increasing numbers of users provide more security in blockchain-based systems. In our system, requesters will be rewarded if their report is confirmed.

## 7. Conclusions

This article proposed a spatial crowdsourcing system designed to increase security and transparency based on an implemented blockchain. Workers are selected according to their location to confirm or reject a report. All users, including the requester, worker, and miner, will receive rewards. These rewards are calculated on the basis of spatial and non-spatial factors which are different for each user. In addition, the proposed system prevents entering incorrect information deliberately or inadvertently, which increases the accuracy of the collected information. Besides, people can monitor how facilities operate. We implemented this system in both Android and iOS platforms. Due to the sensitivity of spatial information, we provided security and privacy analysis. Further, we evaluated the efficiency of this system in the review of received reports by facilities, which decreased by 505.15 min from the minimum review time by reducing 40% of incorrect reports. Future work will focus on the use of smart contracts in this area and the improvement of navigation to facilitate service organizations.

## Figures and Tables

**Figure 1 sensors-21-05146-f001:**
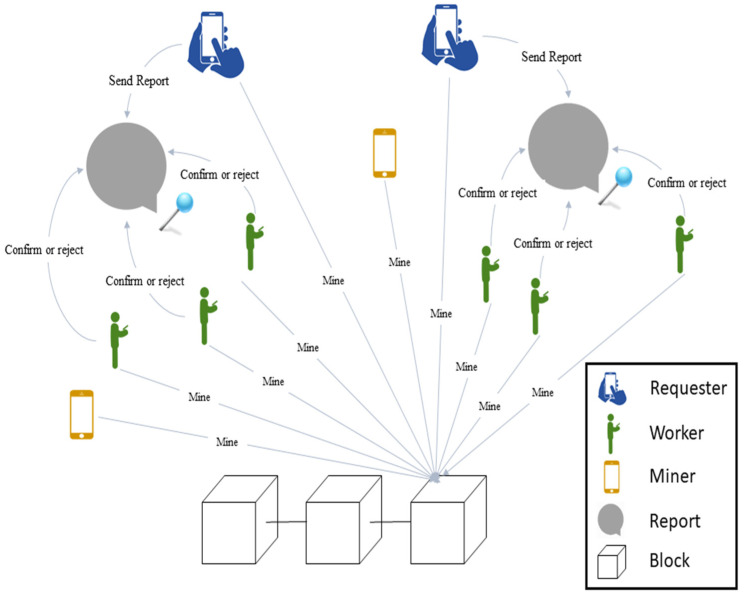
Proposed system architecture.

**Figure 2 sensors-21-05146-f002:**
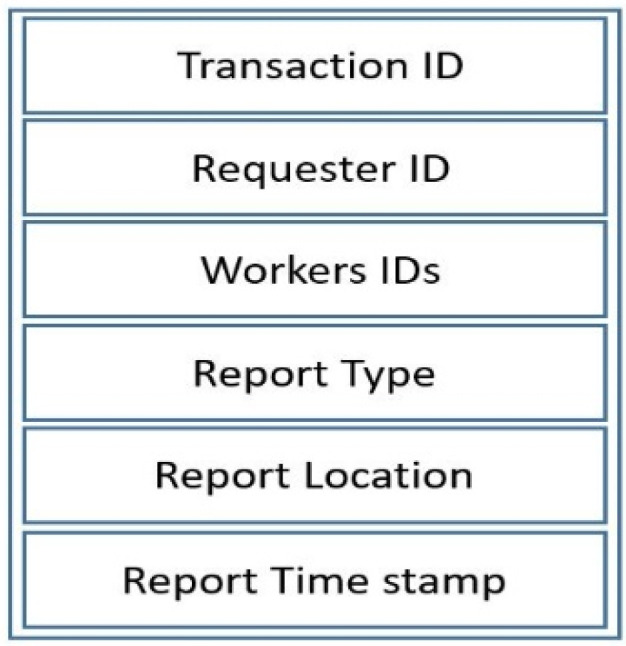
Transaction structure in the proposed system.

**Figure 3 sensors-21-05146-f003:**
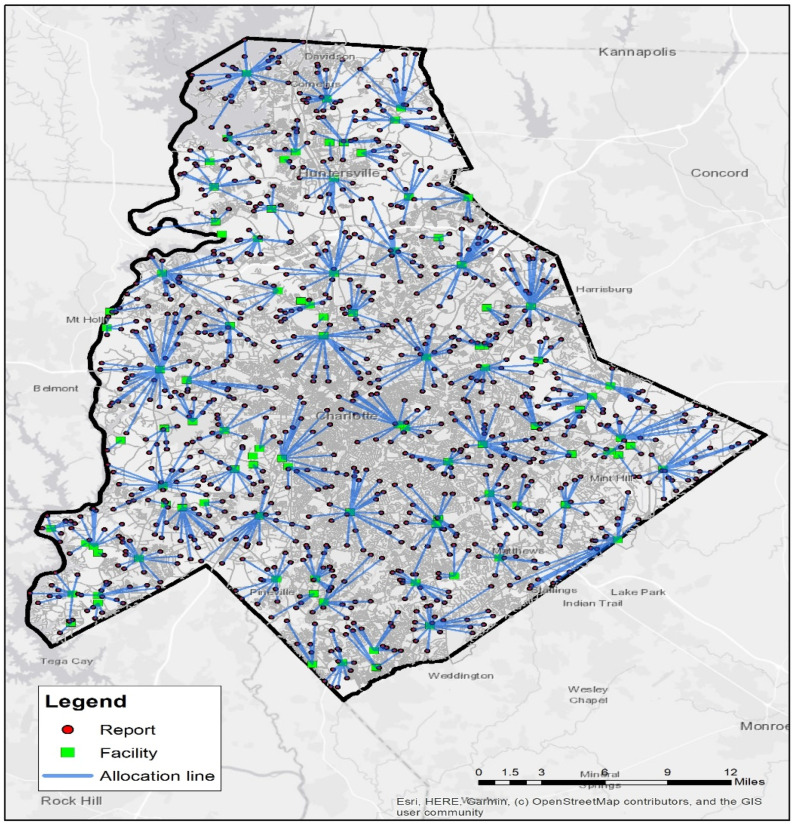
Facility Allocation.

**Figure 4 sensors-21-05146-f004:**
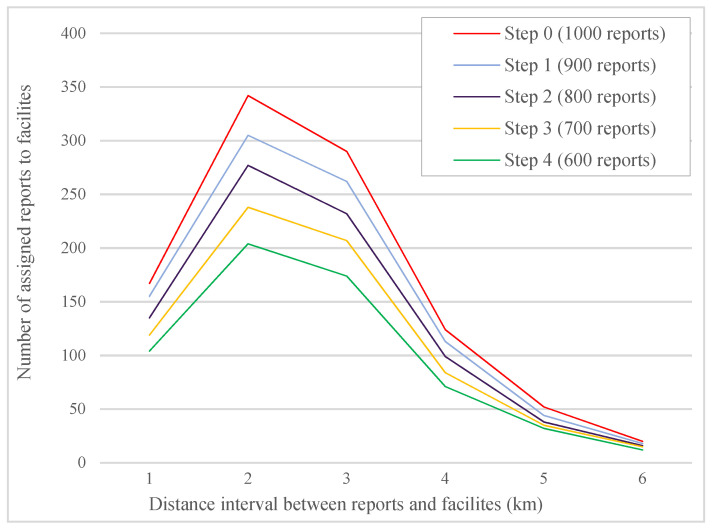
Number of assigned reports to facilities in distance intervals between facilities and reports.

**Figure 5 sensors-21-05146-f005:**
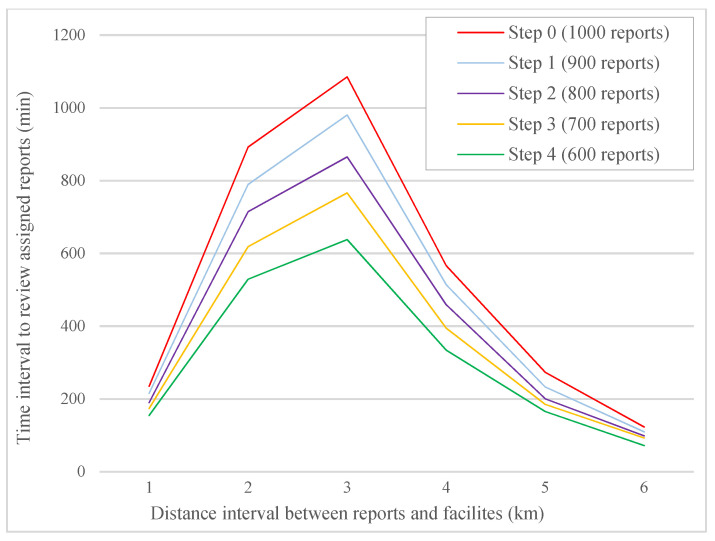
Time interval for facilities to review assigned reports in distance intervals between facilities and reports.

**Table 1 sensors-21-05146-t001:** Total optimal time.

	1000 Reports	900 Reports	800 Reports	700 Reports	600 Reports
Total optimal time (min)	1651.99	1573.74	1329.33	1238.65	1146.84

**Table 2 sensors-21-05146-t002:** Comparison of proposed system with existing crowdsourcing systems.

	Privacy	Worker Credential	Worker Experience	Requester Reputation	Requester Reward
MTurk [58]	✘	✓	✓	✓	✘
CrowdFlower [84]	✘	✓	✓	✓	✘
Jabberwocky [85]	✘	✓	✘	✓	✘
Upwork [86]	✘	✓	✓	✓	✘
CrowdBC [28]	✓	✘	✓	✘	✘
MCS-Chain [81]	✓	✘	✓	✘	✘
Proposed system	✓	✘	✘	✘	✓

## Data Availability

The dataset involved in this paper is public.

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
