# Peer review of "A Blockchain-Based Spatial Crowdsourcing System for Spatial Information Collection Using a Reward Distribution"

_sensors, 2021, doi:10.3390/s21155146_

Round 1

Reviewer 1 Report

Issue 1

The abstract says "...most centralized crowdsourcing systems lack security.." which is too much of  a generalization unless there is a survey or statistics to support the assertion. I recommend replacing 'most' with 'many'.

Issue 2

Line 47: "In this paper, based on the blockchain technology in spatial crowdsourcing" implies that spatial crowdsourcing always has blockchain technology. However, not all spatial crowdsourcing has blockchain technology. Therefore, I recommend revising it to say "In this paper, based on the use of blockchain technology in spatial crowdsourcing".

Issue 3

Line 92 says "According to the different architectures used in the design of crowdsourcing systems, we have classified them into three."

Then Line 93 says "The first category is..."

Line Line 97 says "The second category is..."

I could not find the third category.

Issue 4

Line 137 uses the acronym PPGIS but the acronym has not been introduced in expanded form.

Issue 5

The second paragraph of Section 2 is too long. It goes from Line 88 to 172, which makes it difficult to follow. The paragraph needs to be split into multiple paragraphs.

Issue 6

Line 361 says "We used the Haversine formula to calculate the distance", however there is no citation to the source of the Haversine formula.

Issue 7

Line 417 says "Byzantine fault tolerance" , however there is no citation to the source of the definition of the Byzantine fault tolerance.

Reviewer 2 Report

In the manuscript, the authors have presented the blockchain-based spatial crowdsourcing system in which workers confirm or reject the accuracy of the solved tasks. The topic is interesting, it is structured correctly and contains the necessary sections for this type of publication.

However, to my best mind, there are some shortcomings. Below, I present the remarks which to my mind should be reflected in the manuscript.

  1. The introduction and Related works sections are done qualitatively. However, it will be better if at the end of these sections to allocate the unsolved part of the general problem.
  2. The authors present the results of the proposed model by evaluating its effectiveness and accuracy in comparison with existing models of this type. I think it is necessary to provide a more detailed description of the criteria that were used for this purpose.

Reviewer 3 Report

Authors in this article propose a A Blockchain-based Spatial Crowdsourcing System for Spatial  Information Collection Using a Reward Distribution mechanism. The proposed system design brings certain novel  ideas  in existing work especially ensure security risks minimization through the blockchain based design.  I liked the way authors presented formal description of the system model.

Overall it is very well written paper and could be considered for publication. However, authors need to address following comments to further improve the quality of the paper:

  • The proposed System Architecture diagram (Figure 1) is a bit blur. Please improve its readability by redesigning it.
  • please improve the caption text with meaningful description of Table 1.
  • Authors in section 3.1.2 talk about Thread Model. It is not clear if they are following any threat model. There are certain threat model reference models/standards which should be followed while establishing the Threat Model.
  • Line 237: Authors presume that blockchain may manipulated. But do not explain how it could be compromised. Moreover depending on the type of blockchain used it will be good to link it to the level of security that a blockchain model provides for an information system such as Crowed Sourcing.
  • Starting from line 239: Authors have given a list of threat events it will be good to present these events list with some categorization of threats.
  • Please see the grammar issues 273.
  • In section 5.1 authors talk about consensus algorithms, but it is not clear which type of algorithms are being mentioned.
  • What is the impact of Byzantine faults on overall system security? 
